# The association between new-use of antipsychotics and all-cause mortality in a cohort of patients with dementia in Argentina

María Noelia Vivacqua[1], Pablo Ignacio Osores[1], Héctor Brienza[1], Tomás Barrera[2], José Luis Faccioli[1], Augusto Ferraris[3]*

1 Department of Psychiatry, Hospital Italiano de Buenos Aires, Buenos Aires, Argentina, 2 Department of Internal Medicine, Hospital Italiano de Buenos Aires, Buenos Aires, Argentina, 3 Department of Epidemiology, School of Public Health, University of Washington, Seattle, Washington, United States of America

* aferra@uw.edu

## Abstract

Multiple studies have associated antipsychotic use with increased mortality among individuals with dementia, but evidence in Latin America remains limited. We conducted a retrospective cohort study among outpatients aged 60 years and older receiving care within a large health maintenance organization in Argentina. Participants underwent cognitive evaluation resulting in dementia diagnosis between January 2017 and December 2021. The main exposure was the new-use of antipsychotics after dementia diagnosis. We fitted multivariable Cox proportional hazards models for the association between new-use of antipsychotics and all-cause mortality to estimate adjusted hazard ratios (aHR) and 95% confidence intervals (CI), adjusted for demographics, behavioral factors, and comorbidities. To evaluate dose-response patterns, we used standardized daily doses of antipsychotics (SDD) measured every three months and modelled using restricted cubic splines. Overall, we included 1,326 patients ≥60 years with a new dementia diagnosis, of whom 325 (25%) started antipsychotic treatment during follow-up. Median follow-up time in the entire cohort was 963 (interquartile range: 452–1,333) days and 184/1,326 (14%) of patients died during follow-up. Overall, female sex was most prevalent, and a quarter of patients had completed their secondary education. New-users of antipsychotics had a higher hazard of all-cause mortality compared with non-users (aHR = 2.66, 95%CI: 1.93, 3.67), after adjusting for potential confounders. In secondary analyses, we found no evidence of higher mortality with increasing cumulative antipsychotic exposure. Compared with non-use, the aHRs were inconsistent across cumulative dose levels: 30 SDD (aHR = 2.51, 95%CI: 1.69, 3.74), 90 SDD (aHR = 3.88, 95%CI: 2.30, 6.53), and 365 SDD (aHR = 2.20, 95%CI: 1.22, 3.96). In conclusion, in this retrospective cohort with predominantly female participants with low educational attainment in Argentina, new-use of antipsychotics was associated with higher risk

**Data availability statement:** The data underlying this study consist of clinical records from Hospital Italiano de Buenos Aires and contain potentially identifiable patient information. For this reason, the datasets cannot be shared publicly. Researchers interested in accessing the data to reproduce or extend our analyses may submit a request to the Comité de Ética de Protocolos de Investigación at Hospital Italiano de Buenos Aires. Data access will require institutional review and the execution of appropriate agreements with the hospital. An adapted version of the analytic code, together with synthetic data that allow full implementation of the workflow, is publicly available in GitHub (https://github.com/a-ferraris/antipsychotics_dementia).

**Funding:** The author(s) received no specific funding for this work.

**Competing interests:** The authors have declared that no competing interests exist.

of all-cause mortality among patients with dementia. These findings highlight the importance of minimizing the use of antipsychotics in patients with dementia when feasible.

## Introduction

Dementia is characterized by the progressive decline of cognitive function with preserved consciousness [1]. Approximately 60% to 90% of patients living with dementia develop neuropsychiatric symptoms such as depression, agitation, anxiety, and sleep disturbances [2]. Antipsychotic medications are commonly used to manage these symptoms, and studies report a prevalence of antipsychotic treatment of 20% to 27% among people living with dementia [3–5]. Considering the frequent use of antipsychotics in dementia care, understanding their safety profile is of high public health relevance.

The efficiency and safety of antipsychotics to manage behavioral symptoms in dementia remain controversial. Some studies reported limited benefits in alleviating psychotic symptoms and agitation [6], while others found minimal or no improvements [7]. Conversely, a growing body of evidence shows that antipsychotics can increase mortality in patients living with dementia. Studies suggest that, in patients with dementia, antipsychotic use may exacerbate neurovascular and neurodegenerative processes, increasing susceptibility to hypoperfusion-related injury and contributing to disease progression through interactions with tau pathology, cerebral amyloid angiopathy and small vessel disease [8]. Clinical evidence from meta-analyses of randomized and observational studies have reported an increased risk of all-cause mortality associated with antipsychotic treatment [7,9,10], potentially driven by events such as stroke [7], pneumonia [11], and falls and fractures [7]. These meta-analyses pooled information from studies conducted predominantly in high-income countries, including Canada, Europe, the United States, Australia and New Zealand, with an underrepresentation of the Latin American region [7,9,10]. In Latin America, dementia prevalence is 8% to 17.1% among adults [12], exceeding estimates from high-income countries. Furthermore, factors such as lower educational attainment, prescribing patterns for other central nervous system-active medications, and a higher burden of cardiovascular disease may exacerbate the risks of antipsychotic treatment in patients living with dementia [13]. Nonetheless, evidence about the impact of antipsychotic use on mortality in patients with dementia from Latin American remains underexplored.

To address this gap in knowledge, we conducted a retrospective cohort study of patients newly diagnosed with dementia in Argentina to examine the association between new-use of antipsychotics and all-cause mortality. Furthermore, we evaluated whether cumulative doses of antipsychotic use were associated with a higher risk of all-cause mortality. We hypothesized that patients who initiated antipsychotic treatment would have higher all-cause mortality compared to those who did not, and that higher cumulative doses of antipsychotics may carry higher risk.

## Materials and methods

### Ethics statement

The present study protocol was approved by the local Institutional Review Board, Comité de Ética de Protocolos de Investigación (CEPI) (registration number: 7049) which determined that the requirement for informed consent was waived. MNV, PIO, and HB had access to identifiable data (neurocognitive tests); the author conducting statistical analyses (AF) used limited datasets with potentially identifiable information. Data for the development of the dataset was requested on June 26, 2024, and the initial analysis was conducted on February 15, 2025. The study was conducted in accordance with the principles stated in the 2013 amended Declaration of Helsinki and Good Clinical Practice guidelines [14].

### Data source

We conducted a retrospective cohort study of older adults with a new diagnosis of dementia from January 2017 to December 2021 in Buenos Aires, Argentina. We used individual patient-level data from the Hospital Italiano de Buenos Aires' health maintenance organization to obtain clinical diagnoses, neurocognitive assessments results, and pharmacy dispensing data of older adults with dementia. The Hospital Italiano de Buenos Aires operates a private, not-for-profit healthcare network, serving over 170,000 affiliates in Argentina and it is accredited by the Joint Commission International [15]. The healthcare organization includes a large network of pharmacies, outpatient services, and facilities ranging from low to high complexity, such as two university teaching hospitals with 700 and 125 beds, and numerous medical offices [15].

We captured data on both out-of-hospital and in-hospital diagnoses and procedures, coded using SNOMED CT, Spanish 2020 edition for Windows (SNOMED International, London, UK) [16]. Healthcare professionals enter clinical terms which are subsequently encoded by SNOMED CT using standardized terminology. The procedures to identify and calculate the positive predictive value of each one of the comorbidities included in this study have been previously described [17], where internal medicine specialists conducted manual chart reviews to validate diagnoses. Furthermore, we obtained data on neurocognitive assessments conducted by neuropsychologists and psychiatrists in the outpatient setting. Neurocognitive assessments compile patients' self-reported information, including sociodemographic information (e.g., education), cognitive performance (e.g., Mini Mental State Examination [MMSE] scores), current medication use (e.g., antipsychotic medication), and behavioral (e.g., smoking) and clinical factors (e.g., depressive symptoms scales). Neurocognitive assessments are conducted at the request of clinicians within the healthcare network, and they may be prompted by a variety of reasons such as cognitive complaints or suspected dementia. Finally, we obtained pharmacy dispensing data from the health maintenance organization's pharmacy registry. The organization keeps a comprehensive record of medication purchases by patients through its own network of pharmacies, including medication dose and route of administration. However, the database does not record the number of days of supply or the clinical indication for the filled prescription.

### Study population

We included individuals aged 60 years and older who were members of the Health Maintenance Organization and who underwent an ambulatory neurocognitive assessment resulting in dementia diagnosis between January 1, 2017, and December 31, 2021 (S1 Fig). We defined dementia as a documented diagnosis of dementia in the neurocognitive test (S1 Text) or a score below the age- and education-adjusted cut-off scores on the MMSE (S1 Table). The Spanish-language version of the MMSE has been validated in a previous study in Argentina, using a representative sample of 634 participants aged 30–94 years, stratified sex and educational level [18]. Evidence from other Latin American populations, including Chile, provides further support for the regional generalizability of this screening tool [19]. We excluded patients who filled a prescription for antipsychotics within 90 days prior to the dementia diagnosis (prevalent users) or those with documented use of antipsychotics at the time of diagnosis as reported in the neurocognitive test (S1 Fig) [20].

## Outcome

The main outcome was all-cause mortality during follow-up. Death was captured using the health maintenance organization administrative database, which includes deaths occurring both in the ambulatory and in hospital setting. We followed patients from the date of dementia diagnosis to the earliest of all-cause mortality, five years of follow-up, the end of the study period (i.e., December 31, 2021), or disenrollment from the health maintenance organization (i.e., loss to follow-up).

## Exposure

The main exposure of interest was the new-use of antipsychotics during follow-up. We defined new-use of antipsychotics as filling at least one prescription for antipsychotic medications (e.g., haloperidol, quetiapine) during follow-up (S2 Table). Patients could contribute to both exposed and unexposed person-time at risk, but once patients were exposed to antipsychotics, subsequent changes in the antipsychotic exposure status were disregarded (i.e., "ever-exposed") [21] (S1 Fig). In addition, we calculated the cumulative antipsychotic dose as the sum of standardized daily doses (SDD) present in filled prescriptions during follow-up [22]. SDD represent the average or recommended daily amount of medication per day which allows for the standardized comparison of exposures to different antipsychotics [22]. We updated cumulative antipsychotic exposure every three months. Since lower doses are recommended for the treatment of older adults, we used minimum geriatric doses of antipsychotics to calculate SDD [23]. Minimum geriatric doses reflect routine geriatric prescribing practice, whereby older adults are treated with lower doses than younger populations to achieve clinical response while potentially reducing adverse events, as outlined in a renowned pharmacy reference (S2 Table) [23]. Although SDD do not measure duration of exposure, they can provide clinically meaningful interpretation to cumulative antipsychotic exposure, and they have been used in several studies previously [24–27]. For example, a filled prescription for 30 tablets of quetiapine 25 mg corresponds to a cumulative dose of 750 mg (30 times 25). This quantity is then divided by the recommended dose for quetiapine in older adults (12.5 mg), yielding 60 SDD. Hence, the filled prescription for quetiapine equals a cumulative dose of 60 SDD, which can be interpreted as the equivalent of 60 days of supply, assuming the recommended dose was taken each day.

## Other covariates

We captured sociodemographic information, neurocognitive scores, pharmacy dispensing data, and comorbidities. Neurocognitive tests were used to retrieve data on age at time of diagnosis, sex (female or male), MMSE scores (0–30, assessed by expert personnel: lower scores imply worse cognitive performances), Beck Depression Inventory II scores (0–63, self-reported: highest scores imply more severe symptoms), and highest education level achieved (self-reported or reported by caregiver: less than primary, primary complete, secondary complete, or higher than secondary). We used pharmacy dispensing data to identify filled prescriptions for opioids, antidepressants, antiepileptics, benzodiazepines, and z-drugs within 90 days prior to the dementia diagnosis. Clinical diagnoses were used to obtain information on Parkinson disease, Lewy body dementia, tobacco use, epilepsy, asthma, hypertension, chronic kidney disease, diabetes mellitus, insomnia, depressive disorders, anxiety, peripheral vascular disease, cardiac failure, and coronary disease.

## Statistical analysis

We handled missing data using multiple imputation by chained equations. Missing data were present in education (1.0%), Beck Depression Inventory II scores (15.0%), and MMSE scores (1.1%). We imputed 10 datasets using a maximum of 50 iterations, and calculated pooled estimates and confidence intervals following Rubin's rules [28]. The imputation model included only variables that were also used in the analysis model (see below), and variables were selected iteratively until convergence was achieved. We used data on age, sex, cancer, diabetes mellitus, chronic obstructive pulmonary disease, insomnia, benzodiazepine, opioid, z-drug, antidepressant and antiepileptic drug use at baseline to impute values on variables with missing information using random forests [29].

We used multivariable cox proportional hazards models to calculate adjusted hazard ratios (aHR) and 95% confidence intervals (CI). In our main analysis, we evaluated the association between the new-use of antipsychotics and all-cause mortality over five years of follow-up. Censoring was assumed to be independent (i.e., the mechanism of censoring is non-informative). Covariates were selected based on prior subject matter knowledge [30]. We adjusted for Beck Depression Inventory II score (continuous, linear), MMSE score (continuous, linear), highest education degree attained (indicator variable coding), age (continuous, linear), and as binary variables: sex, Parkinson disease, Lewy body dementia, tobacco use, epilepsy, asthma, hypertension, chronic kidney disease, diabetes mellitus, insomnia, depressive disorders, anxiety, peripheral vascular disease, cardiac failure, coronary disease, cancer, chronic obstructive pulmonary disease, and opioid, antidepressant, benzodiazepine, z-drugs, and antiepileptic medication use. We updated every three months information about insomnia, depressive disorders, anxiety, cardiac failure, coronary disease, and opioid, antidepressant, benzodiazepine, z-drug, and antiepileptic use (i.e., time-varying) (S1 Text).

### Secondary and sensitivity analyses

Secondary analyses evaluated cumulative antipsychotic use and comparisons for typical and atypical antipsychotics. To evaluate whether higher cumulative antipsychotic doses were associated with a higher mortality risk, we conducted an exploratory secondary dose-response analysis. We used restricted cubic splines to model the cumulative exposure to antipsychotics, updated every three months. Splines are curves which allow for the flexible model of continuous variables without assuming a fixed a priori shape [31]. To account for the potential of reverse causation (e.g., dementia progression prompting antipsychotic use), we incorporated lag time periods in this analysis, in which cumulative antipsychotic use was delayed three and six months [32]. Hence, antipsychotic use in the three or six months preceding an event would not be accounted for by the calculation of cumulative antipsychotic use.

We performed sensitivity analyses to assess the potential impact of study design features on the estimates. First, the risk of all-cause mortality may be present early in the treatment with antipsychotics [10]. We re-fitted the main model modifying patient's follow-up to one and three years. Second, we changed our definition of prevalent user to exclude patients who filled prescriptions for antipsychotics within the 90 days preceding dementia diagnosis to 180 and 365 days. Third, to evaluate the impact of our assumption of a fixed exposure to antipsychotics (i.e., "ever exposed") in our results, which can lead to exposure misclassification [20], we repeated the main model but changed modelling of the main exposure, allowing patients to start and stop antipsychotic treatment. We created exposure intervals, defined by a period of 90 days after a filled prescription for an antipsychotic. Hence, we considered patients exposed to antipsychotics for 90 days after a prescription for antipsychotics was filled, accounting for overlapping time periods of exposure. Finally, we repeated the same procedure but changing the duration of exposure intervals to 180 days.

All analyses were conducted using R statistical software version 4.4.1 (R Foundation for Statistical Computing, Vienna, Austria) [33]. We used a threshold of .05 to define statistical significance in all tests. This manuscript follows the REporting of studies Conducted using Observational Routinely collected health Data statement for PharmacoEpidemiology (RECORD-PE) statement [34].

### Results

We included a total of 1,326 patients ≥60 years with a new dementia diagnosis between January 1, 2017, and December 31, 2021 (Fig 1). Female sex was most prevalent and approximately a quarter of patients had completed their secondary education. Antidepressants and benzodiazepines were the most frequently prescribed central nervous system-active medications at the time of dementia diagnosis (Table 1). Overall, 325/1,326 (25%) patients started treatment with an antipsychotic during follow-up. On average, patients who started antipsychotics were older, more likely to be female, and had a lower education level at baseline, compared with those who did not start treatment during follow-up (Table 1). Moreover, new users of antipsychotics had on average lower MMSE, higher Beck Depression Inventory II scores and were more

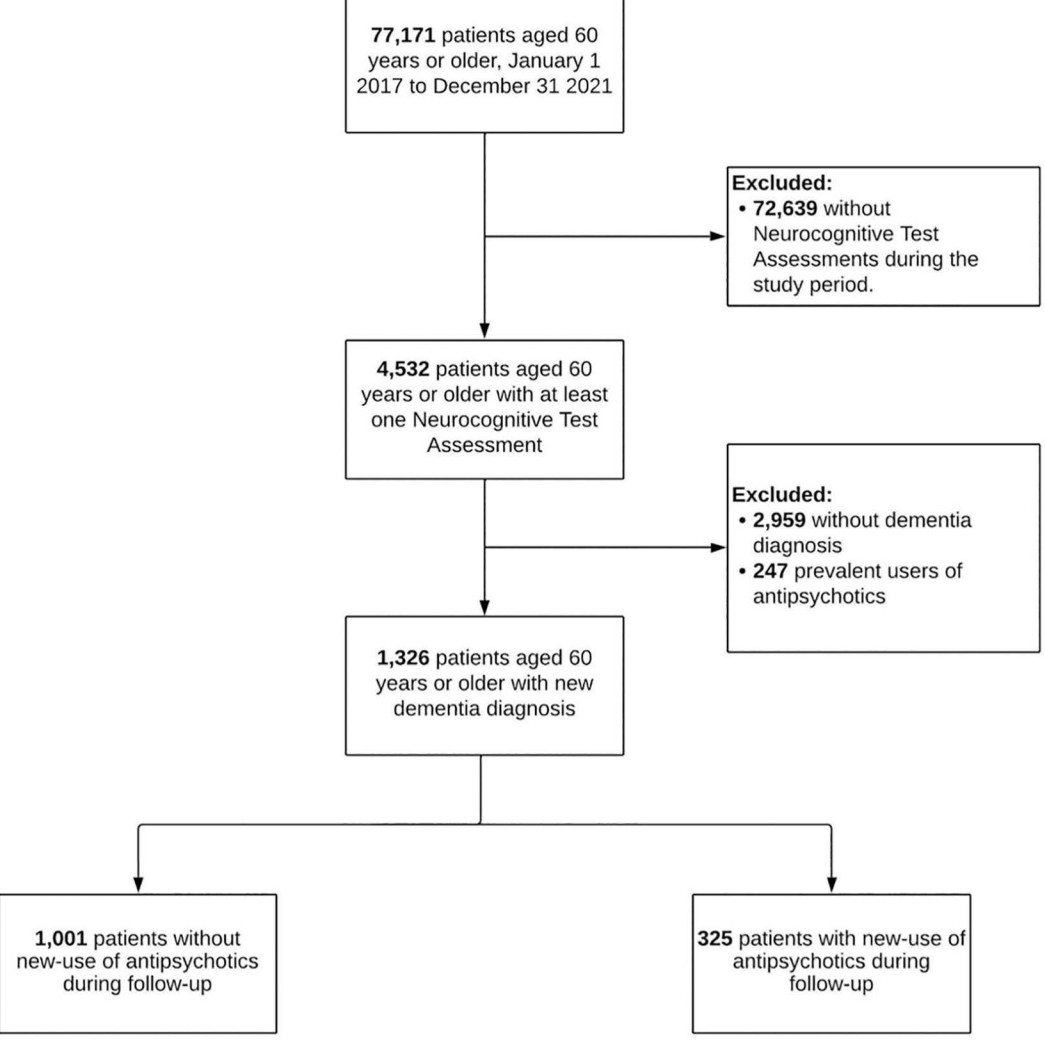

**Fig 1. Flowchart of study participants.**

likely to be treated with benzodiazepines, antidepressants, and antiepileptics at baseline. Patients treated with antipsychotics had a higher burden of cardiac failure, cancer, epilepsy, insomnia, and chronic obstructive pulmonary disease, but were less likely to have diabetes mellitus and coronary disease (Table 1). Finally, the use of central nervous system-active medications were the covariates to change the most during follow-up, while clinical comorbidities remained mostly unchanged, on average (S3 Table).

Median follow-up time in the entire cohort was 963 days (interquartile range: 452–1,333). A total of 184/1,326 (14%) patients died, and 38/1,326 (3%) patients disaffiliated from the health maintenance organization during follow-up. The median time until the first antipsychotic prescription was 344 days (interquartile range: 114–698). Atypical antipsychotics (e.g., quetiapine) were used in 93% of patients who started antipsychotic treatment. The most frequent antipsychotic was quetiapine (75% of users), followed by risperidone (14%), olanzapine (4%), levomepromazine (3%), haloperidol (1%), aripiprazole (1%), and others (1%). The median cumulative antipsychotic exposure was 360 SDD (interquartile range: 120, 1080).

**Table 1. Baseline characteristics of patients with a diagnosis of dementia at Hospital Italiano de Buenos Aires by antipsychotic drug use during follow-up, 2017-2021.**

| Baseline characteristics | Antipsychotic use (N = 325) | No antipsychotic use (N = 1,001) |
|---|---|---|
| Age, mean (SD) | 80.7 (7.0) | 76.8 (7.4) |
| Female, n (%) | 236 (73) | 653 (65) |
| Education, n (%) | | |
| Less than primary, | 32 (10) | 72 (7) |
| Primary complete | 235 (73) | 674 (67) |
| Secondary complete | 49 (15) | 220 (22) |
| Tertiary or higher | <10 (<3) | 35 (4) |
| Tobacco use, n (%) | 57 (18) | 182 (18) |
| **Mental health conditions, n (%)** | | |
| BDI score, mean (SD) | 16.15 (9.96) | 12.80 (9.23) |
| MMSE score, mean (SD) | 20.82 (4.62) | 22.96 (3.82) |
| Parkinson's disease | 13 (4) | 31 (3) |
| Lewy body | 54 (17) | 159 (16) |
| Insomnia | 47 (15) | 124 (12) |
| **Other comorbidities, n (%)** | | |
| Asthma | 20 (6) | 59 (6) |
| Cancer | 47 (15) | 114 (11) |
| Cardiac failure | 21 (7) | 50 (5) |
| Chronic kidney disease | 25 (8) | 59 (6) |
| COPD | 29 (9) | 47 (5) |
| Coronary disease | 32 (10) | 121 (12) |
| Diabetes mellitus | 39 (12) | 162 (16) |
| Epilepsy | 10 (3) | 12 (1) |
| Hypertension | 228 (70) | 731 (73) |
| Peripheral vascular disease | 10 (3) | 30 (3) |
| **Medications, n (%)** | | |
| Opioids | 11 (3) | 22 (2) |
| Benzodiazepines | 74 (23) | 135 (14) |
| Antidepressants | 69 (21) | 153 (15) |
| Antiepileptics | 22 (7) | 42 (4) |
| Z drugs | <10 (<3) | 16 (2) |

BDI: Beck Depression Index II, COPD: chronic obstructive pulmonary disease, MMSE: Mini mental state examination.

The crude incidence rate of all-cause mortality was 17.1 per 100 person-years and 3.5 per 100 person-years in the antipsychotic use and non-use groups, respectively. In the main model, new-use of antipsychotics was associated with 2.66 times (95%CI: 1.93 to 3.67) higher hazard of all-cause mortality compared with non-use, after adjusting for potential confounders (Table 2). In secondary exploratory analysis, cumulative antipsychotic exposure did not show a clear dose-response pattern with all-cause mortality. We observed a marked increase in the aHR of all-cause mortality with low cumulative doses, compared with no use, followed by a decrease in aHR at higher cumulative doses, compared with no use (Fig 2). Estimates for selected cumulative antipsychotic medication use exposures are presented in Table 3.

**Table 2. Results of Cox Proportional hazards models for main, secondary, and sensitivity analysis.**

| Analysis | Number of events, exposed | Person-years, exposed | Number of events, unexposed | Person-years, unexposed | Adjusted Hazard Ratio (95% CI)[1] |
|---|---|---|---|---|---|
| Main model[1] | 83 | 484.0 | 100 | 2835.4 | 2.66 (1.93, 3.67) |
| **Sensitivity analysis** | | | | | |
| Follow-up, 1 year | 15 | 90.3 | 27 | 1091.4 | 1.84 (1.23, 2.74) |
| Follow-up, 3 years | 63 | 372.3 | 81 | 2468.0 | 2.64 (1.91, 3.67) |
| 6-months prevalent use[2] | 81 | 465.3 | 100 | 2822.7 | 2.75 (1.99, 3.80) |
| 12-months prevalent use[3] | 79 | 460.2 | 100 | 2809.3 | 2.65 (1.92, 3.68) |
| 3-month exposure definition | 58 | 353.4 | 125 | 2939.4 | 1.99 (1.39, 2.83) |
| 6-month exposure definition | 72 | 391.9 | 111 | 2900.9 | 2.59 (1.84, 3.69) |

CI confidence interval; SDD: standardized daily doses.

1. All models are adjusted for time-fixed: education, Beck Depression Inventory II score (linear continuous), education (less than primary, primary complete, secondary complete, higher than secondary), age (linear), female sex, Parkinson disease, frontotemporal, vascular, mixed, Alzheimer's disease, and Lewy body dementia, Mini mental examination score, tobacco use, alcohol use, bipolar disease, epilepsy, asthma, hypertension, chronic kidney disease, diabetes mellitus, chronic obstructive pulmonary disease, cancer; and as time-varying confounders: insomnia, depressive disorders, anxiety, cardiac failure, coronary disease, opioid use, antidepressant use, benzodiazepine use, z drug use, and antiepileptic drug use.

2. A total of 1313 patients included.

3. A total of 1304 patients included.

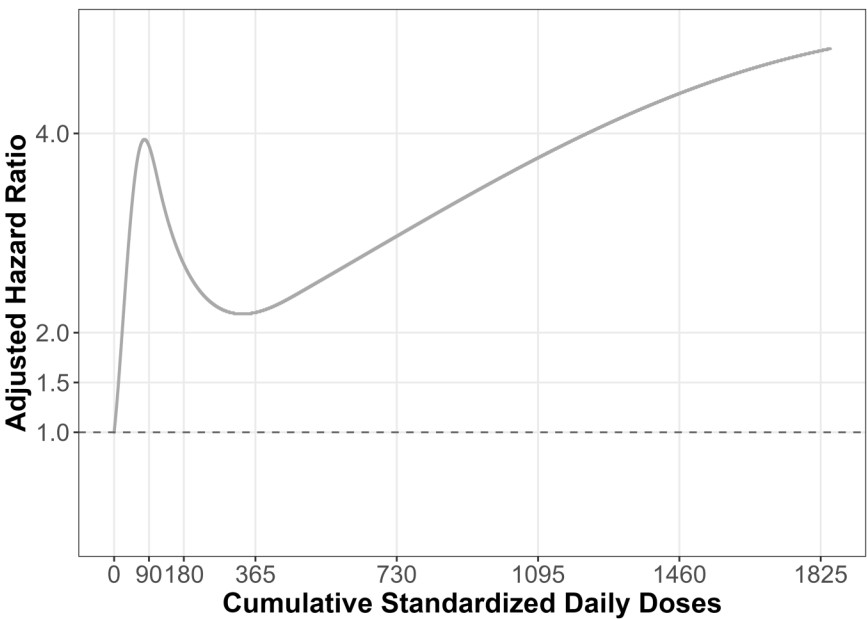

**Fig 2. Cumulative antipsychotic doses and risk of all-cause mortality in secondary exploratory models.**

## Discussion

We evaluated the association between new-use of antipsychotics and all-cause mortality among older adults with a new dementia diagnosis in Argentina. We found a higher risk of all-cause mortality after adjusting for sociodemographic, behavioral, and clinical variables. Results remained consistent across sensitivity analysis, modifying study design

**Table 3. Results of Cox Proportional hazards models for cumulative antipsychotic exposure using restricted cubic splines with lag periods.**

| Exposure level | 3-months lag period, Adjusted Hazard Ratio (95% CI)[1] | 6-months lag period, Adjusted Hazard Ratio (95% CI)[1] |
|---|---|---|
| 0 SDD | Ref. | Ref. |
| 30 SDD | 2.51 (1.69, 3.74) | 2.92 (1.97, 4.31) |
| 90 SDD | 3.88 (2.30, 6.53) | 3.97 (2.40, 6.57) |
| 180 SDD | 2.69 (1.68, 4.30) | 1.91 (1.11, 3.29) |
| 365 SDD | 2.20 (1.22, 3.96) | 1.40 (0.72, 2.71) |
| 1095 SDD | 3.76 (2.23, 6.31) | 3.36 (1.89, 6.00) |
| 1825 SDD | 4.83 (2.46, 9.49) | 4.89 (2.33, 10.27) |

CI: confidence intervals, SDD: standardized Daily doses

procedures and exposure modelling. In secondary exploratory analysis, we did not find evidence of a clear dose-response pattern for cumulative antipsychotic exposure.

This study adds to the body of evidence showing that antipsychotic use is associated with a higher risk of all-cause mortality in patients with dementia. Prior research has reported estimates compatible with effect sizes ranging from a 1.06- to 2.23-fold higher risk of mortality in antipsychotic users compared to non-users [7,10,35]. In our main analysis, we found an association lying in the upper end of such range, which may be attributed to several potential factors. First, evidence suggests that antipsychotic use could affect dementia trajectories, accelerating tau protein-related pathologic processes, increasing patients susceptibility to cerebral hypoperfusion from sedation-related events, and producing a greater burden of small vessel disease and cerebral amyloid angiopathy [8]. Nonetheless, we lacked information on surrogates of such potential pathways. Second, a high baseline risk of all-cause mortality in our study population may explain our findings. This study included predominantly female patients, who face specific challenges as older adults. As in many aging populations, older adults in Argentina often experience progressive loss of functional independence and social support networks, due to biological changes and life-course transitions (e.g., retirement). Previous studies suggest that demographic shifts (e.g., declining birth rates) may have reduced the availability of informal caregiving, potentially increasing caregiver burden [36,37]. In resource-constrained settings like Argentina, these circumstances are frequently accompanied by economic vulnerability. Specifically, among women, lifelong gender inequities (e.g., lower education, wage gaps) have led to precarious retirement conditions and greater economic dependency on male partners [38]. Collectively, these factors may contribute to insufficient care and limited access to non-pharmacological interventions. Hence, the risk of all-cause mortality associated with antipsychotic use may be exacerbated by the distribution of social determinants of health in our study population. Third, our study period included the onset of the COVID-19 pandemic in Argentina, where one of the longest mandatory lockdowns worldwide took place [39]. In the Buenos Aires Metropolitan Area, were most participants of this study potentially resided [15], the country implemented the strictest lockdown and social distancing measures [40]. These restrictions may have had an even greater impact on older adults receiving antipsychotic treatment, who are particularly vulnerable to disruptions in care [41].

This study has several limitations. First, we used a retrospective design, and we relied on routinely collected data to identify new dementia cases. However, we leveraged standardized assessments to identify dementia cases and mental health symptoms, including MMSE scores and depressive symptoms scales. Second, as previously mentioned, unmeasured and residual confounding remain a potential source of bias. For example, Beck Depressive Inventory scores are obtained by self-report, and there is potential for inaccuracy in its measurement, especially among patients living with

dementia [42]. Third, we ascertained antipsychotic exposure using pharmacy dispensing data and misclassification of exposure status is possible. Nonetheless, filled prescriptions are considered the gold standard for exposure ascertainment in most pharmacoepidemiologic research [43]. In addition, the risk of potential misclassification of exposure status may be low in this specific setting. In Argentina, antipsychotic medications require duplicated prescriptions to ensure their traceability, and patients receive discounts when acquiring medications within the health maintenance organization pharmacy network [15]. Fourth, due to limitations related to sample size, we could not explore the associations of individual antipsychotic medications, or medication classes (i.e., "typical" or "atypical") and all-cause mortality. Our study had a predominant representation of quetiapine and risperidone users, and other antipsychotics represented a fraction of the exposed individuals. Nevertheless, these patterns of antipsychotic use may reflect those of the real-world in the setting of Argentina [17,44]. Finally, participants of this study were affiliated with a health maintenance organization and underwent cognitive testing for dementia. By requiring a cognitive test for inclusion, the study may have disproportionately enrolled individuals with more prominent symptoms, better healthcare access, or distinct health-seeking patterns, either alone or in combination [45]. This may select patients who are not representative of the broader population of individuals living with dementia [45]. In particular, our study population consisted largely of urban, middle-class adults, and the findings may not be generalized to rural populations or to groups with different socioeconomic characteristics or healthcare seeking behaviors.

This study has strengths. First, this study contributes to dementia care evidence in Argentina and Latin America, where the access to healthcare services, sociodemographic characteristics and prescribing practices largely differ from high-income countries [46]. Second, we used information identified in standardized clinical tests to ascertain dementia diagnosis and severity, avoiding misclassification of dementia status. Third, we captured both short- and long-term mortality, and we included sensitivity analysis considering varying follow-up times. These procedures are relevant given the prolonged use and cumulative risks of antipsychotics in dementia reported in prior studies [47,48]. Finally, we tested for a dose-response pattern. We did not find strong evidence of greater all-cause mortality with higher cumulative doses of antipsychotics, and further studies reproducing our methods are needed to confirm such findings.

## Conclusions

In a cohort of patients with dementia, with predominance of women with low educational attainment in Argentina, we identified a higher risk for all-cause mortality among new users of antipsychotics compared to non-users, after adjusting for sociodemographic, pharmacological, and clinical confounders. This study underscores the need for deprescribing antipsychotic medications when feasible and enhancing the use of non-pharmacological interventions to manage dementia care.

## Supporting information

**S1 Text. Definition of study variables and R packages.** This file includes detailed definitions of dementia-related diagnostic terminology, psychotropic drug exposure, and technical analytical detail.
(DOCX)

**S1 Fig. Graphical representation of the inclusion and exclusion criteria and exposure ascertainment.**
(TIFF)

**S1 Table. MMSE score thresholds adjusted by age and education level.** This table provides the cut-off scores used to identify cognitive impairment using the Mini Mental State Examination (MMSE). Thresholds are adjusted for both age and years of education. These values were applied when an explicit diagnosis of Major Neurocognitive Disorder was not documented in the clinical records.
(DOCX)

**S2 Table. Antipsychotics drugs included and Minimum geriatric doses of antipsychotics.** This table lists standardized daily doses (SDD) for antipsychotic medications in older adults.
(DOCX)

**S3 Table. Potential confounding over time.** This table depicts the proportion of participants with time-varying covariates since dementia diagnosis at different time points during follow-up.
(DOCX)

## Author contributions

**Conceptualization:** María Noelia Vivacqua, Pablo Ignacio Osores, Héctor Brienza, Tomás Barrera, José Luis Faccioli, Augusto Ferraris.

**Data curation:** María Noelia Vivacqua.

**Formal analysis:** Augusto Ferraris.

**Methodology:** María Noelia Vivacqua, Augusto Ferraris.

**Project administration:** María Noelia Vivacqua.

**Supervision:** María Noelia Vivacqua.

**Writing – original draft:** María Noelia Vivacqua.

**Writing – review & editing:** Pablo Ignacio Osores, Héctor Brienza, Tomás Barrera, José Luis Faccioli, Augusto Ferraris.

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
