## [Decision Letter · Decision Letter 0]

22 Oct 2025

PMEN-D-25-00420

The association between antipsychotic initiation and all-cause mortality in a cohort of patients with dementia in Argentina.

PLOS Mental Health

Dear Dr. Ferraris,

Thank you for submitting your manuscript to PLOS Mental Health. After careful consideration of reviewers' feedback, we feel that it has merit but does not fully meet PLOS Mental Health’s publication criteria as it currently stands. Therefore, we invite you to submit a revised version of the manuscript that addresses the points raised during the review process.

We look forward to receiving your revised manuscript.

Kind regards,

Rose Mary Xavier, PhD, APRN, PMHNP-BC, FAAN,FAANP

Academic Editor

PLOS Mental Health

Journal Requirements:

1. We note that you have indicated that there are restrictions to data sharing for this study. For studies involving human research participant data or other sensitive data, we encourage authors to share de-identified or anonymized data. However, when data cannot be publicly shared for ethical reasons, we allow authors to make their data sets available upon request. For information on unacceptable data access restrictions, please see http://journals.plos.org/plosone/s/data-availability#loc-unacceptable-data-access-restrictions.

Additional Editor Comments (if provided):

Reviewers' comments:

Reviewer's Responses to Questions

**Comments to the Author**

1. Does this manuscript meet PLOS Mental Health’s publication criteria?

Reviewer #1: Yes

Reviewer #2: Yes

2. Has the statistical analysis been performed appropriately and rigorously?

Reviewer #1: Yes

Reviewer #2: I don't know

3. Have the authors made all data underlying the findings in their manuscript fully available (please refer to the Data Availability Statement at the start of the manuscript PDF file)?

Reviewer #1: No

Reviewer #2: Yes

4. Is the manuscript presented in an intelligible fashion and written in standard English?

Reviewer #1: Yes

Reviewer #2: Yes

Reviewer #1: This manuscript makes a valuable contribution to the limited literature on geriatric mental health in low- and middle-income countries by examining the association between antipsychotic initiation and all-cause mortality in a dementia cohort in Argentina. It is a relevant and timely topic. The work is generally methodologically sound, and the statistical approach seems appropriate, although reproducibility could be enhanced by providing access to the dataset and R code.

The following are minor questions and suggestions to help clarify aspects of the manuscript:

DATA SOURCE

LINE 137: The authors did well to describe SDD, but it is not immediately clear how it was calculated in this study. If the number of days of supply of antipsychotics was missing from the pharmacy data, did the authors estimate from dosage instructions, or infer from refill intervals, etc.?

STUDY POPULATION

LINE 140: Epidemiological studies on dementia in Latin America seem to favor the age of 65 years as a cutoff (Table 3, PMID: 29305437). It would be beneficial to clarify the rationale behind the authors' choice of 60 years in this study.

LINE 143: Since the authors used MMSE as a proxy for dementia in certain cases, it would strengthen the manuscript to include information about validation studies of this screening tool on the Latin American population at a minimum, or in Argentina if available.

OUTCOME

LINE 152: How different is “five years of follow-up” from “the end of the study period”? If there is a difference, when did the study period end?

LINE 153: What does "disenrollment" from the HMO mean? Does it imply a loss to follow-up?

SECONDARY AND SENSITIVITY ANALYSES

LINE 235: In the interest of reproducibility, can the R codes and packages used in the analysis be included in the supplementary table?

- A brief comment on how censoring was handled would improve clarity.

DISCUSSION

LINE 351: I suggest rephrasing the latter portion of this sentence because, although MMSE is widely used to stage dementia severity, it is technically a screening tool for cognitive impairment.

- A brief statement on the generalizability of these findings would be helpful.

In summary, this is a well-conceived and relevant study. Addressing the points above would further strengthen the manuscript's impact.

Reviewer #2: This manuscript addresses antipsychotic safety in dementia patients. It examines mortality risk in older adults newly diagnosed with dementia in Argentina and finds an increased risk of death among antipsychotic users. With some revisions, this work would make a novel and important contribution to the literature.

-Consider specifying "new-use" in the title

-In the abstract, confirm if it is okay to specify the exact location and name of the hospital that patients came from. “outpatients aged 60 years and older at the Hospital Italiano de Buenos Aires’ health maintenance organization in Argentina” could potentially constitute identifying information

-line 83: it may be good to provide a range instead of an exact figure, as I’ve seen higher estimates: “dementia prevalence is 7% among adults aged ≥75 years"

-in the introduction, consider discussing specific mechanisms linking antipsychotics to early mortality, if possible

-in the methods, consider discussing potential selection bias from requiring individuals who have “underwent an ambulatory neurocognitive assessment” (Line 141), even if it may be a prerequisite for dementia diagnosis

-Line 167: in the methods, the choice of minimum geriatric doses may need better justification and include what this dosage is or the range it may constitute. It may also be good to specify what this “clinical practice” is and what standards are used to determine this minimum dosage

-Line 188: For statistical analysis, I find there to be fairly sparse imputation details with just one sentence saying: ‘We handled missing data using multiple imputation by chained equations.”. Which variables had missing data and why were specific predictors chosen?

-in the results, the manuscript says, “In addition, we observed a higher risk of all-cause mortality among patients treated with typical antipsychotics (aHR = 4.41, 95% CI: 2.06 to 9.44) compared with non-users, and this risk was greater than that associated with atypical antipsychotics (aHR = 2.54, 95% CI: 2.80 to 3.58), compared with non-users”. Firstly, it would be good to note the small sample size and consider whether or not to include this statement given the weak power, as the typical antipsychotic group has only 27 person years and seem to potentially be as low as 13 patients? Furthermore, there is a discrepancy between table 2 and what is written in the paragraph. It appears the aHR for atypical antipsychotics should be 2.54 (1.80-3.58), not (2.80-3.58) that is written. Please double check all numbers in the manuscript to ensure alignment with the tables

-at times there seems to be abbreviation inconsistency with regards to “aHR” vs “adjusted hazard ratio”. Please ensure consistency throughout except the first instance, where it is expanded.

-in the discussion, I appreciate the mention of challenges female older adult Argentinians face. That being said, mention of more older-adult specific challenges would be beneficial. For instance, lower functional independence, lack of social supports, etc. These are generally well recognized in the literature. As it stands, the discussion primarily touches upon covid 19, confounding, and gender inequality. While all important, these appear quite general and could apply to any population of study, not specifically to older adults taking antipsychotics.

-Line 344: ‘Finally, the strong association between antipsychotic use and mortality in this study may reflect unmeasured or residual confounding (36)”. This statement is too general, as all studies have the potential for confounding. It would be good to either remove this sentence if the authors feel like there are not particularly poignant potential confounders readers should know about, or if there are, to move this to the limitations where it is discussed in more detail

**Do you want your identity to be public for this peer review?** For information about this choice, including consent withdrawal, please see our Privacy Policy

Reviewer #1: No

Reviewer #2: **Yes:** Sunny Cui

---

## [Editor Report · Decision Letter 1]

28 Jan 2026

The association between new-use of antipsychotics and all-cause mortality in a cohort of patients with dementia in Argentina.

PMEN-D-25-00420R1

Dear Dr. Ferraris,

We are pleased to inform you that your manuscript 'The association between new-use of antipsychotics and all-cause mortality in a cohort of patients with dementia in Argentina.' has been provisionally accepted for publication in PLOS Mental Health.

Best regards,

Rose Mary Xavier, PhD, APRN, PMHNP-BC, FAAN,FAANP

Academic Editor

PLOS Mental Health